# Metabolic Reprogramming of Innate Immune Cells as a Possible Source of New Therapeutic Approaches in Autoimmunity

**DOI:** 10.3390/cells11101663

**Published:** 2022-05-17

**Authors:** Leila Mohammadnezhad, Mojtaba Shekarkar Azgomi, Marco Pio La Manna, Guido Sireci, Chiara Rizzo, Giusto Davide Badami, Bartolo Tamburini, Francesco Dieli, Giuliana Guggino, Nadia Caccamo

**Affiliations:** 1Central Laboratory of Advanced Diagnosis and Biomedical Research (CLADIBIOR), 90127 Palermo, Italy; leila.mohammadnezhad@unipa.it (L.M.); mojtaba.shekarkarazgomi@unipa.it (M.S.A.); guido.sireci@unipa.it (G.S.); giustodavide.badami@unipa.it (G.D.B.); bartolo.tamburini@unipa.it (B.T.); francesco.dieli@unipa.it (F.D.); nadia.caccamo@unipa.it (N.C.); 2Department of Biomedicine, Neurosciences and Advanced Diagnostic (Bi.N.D.), University of Palermo, 90127 Palermo, Italy; 3Rheumatology Section, Internal Medicine and Medical Specialties, Department of Health Promotion, Mother and Child Care, University of Palermo, Piazza delle Cliniche 2, 90110 Palermo, Italy; chiara.rizzo06@you.unipa.it (C.R.); giuliana.guggino@unipa.it (G.G.)

**Keywords:** chronic inflammatory disease, autoimmunity, innate immunity, immune response, therapy, metabolic pathways, glycolysis, oxidative phosphorylation, HIF-1α

## Abstract

Immune cells undergo different metabolic pathways or immunometabolisms to interact with various antigens. Immunometabolism links immunological and metabolic processes and is critical for innate and adaptive immunity. Although metabolic reprogramming is necessary for cell differentiation and proliferation, it may mediate the imbalance of immune homeostasis, leading to the pathogenesis and development of some diseases, such as autoimmune diseases. Here, we discuss the effects of metabolic changes in autoimmune diseases, exerted by the leading actors of innate immunity, and their role in autoimmunity pathogenesis, suggesting many immunotherapeutic approaches.

## 1. Introduction

The immune system consists of different cells, tissues, and organs that have the ability to respond to self-endogenous stimuli and non-self-exogenous pathogens. A specific response is induced by the adaptive immune system, and a unspecific and fast response is exerted by innate immune cells [1,2]. Innate immunity is made up of cells such as monocytes, macrophages, dendritic cells, neutrophils, mast cells, and natural killer (NK) cells [1]. Immune cells are plastic, and following a stimuli, these cells switch their metabolic program to be activated and defend tissue. Cells changing their metabolic program requires much more energy, which can be provided through oxidative phosphorylation (OXPHOS) and glycolysis [3], which is discussed more here.

Glycolysis is the metabolic pathway initiated with glucose uptake through glucose transporters (GLUTs). It consists of several steps, whereby briefly glucose is converted to pyruvate by different enzymes. Hexokinase (HK) is involved in the first irreversible step, followed by other products and enzymes. The final product of glycolysis is pyruvate, which then enters into the Krebs cycle (mitochondria) or can be reduced to lactate by lactate dehydrogenase A (LDH-A) [4,5,6,7].

Two ATP for each glucose and NADH are generated during this pathway [8]. Although glycolysis yields less ATP production than OXPHOS, activated cells prefer using glycolysis rather than the mitochondrial tricarboxylic acid (TCA) cycle [9], to provide the high energy demands for rapid cell differentiation and immune responses. The link between immune cells and the metabolic program is called immunometabolism, which determines cells’ survival, function, and differentiation, and thus is under strict control [10,11]. Even though immune metabolism is essential for cells’ survival and differentiation, dysregulation of its controllers or regulators could elicit pathogenicity. A growing body of evidence suggests that innate cells with dysregulated metabolic programs may mediate many inflammatory conditions. Therefore, in this review, we consider the metabolic program of the main subsets of innate immune cells: macrophages, dendritic cells, neutrophils, and the role of these cells in several autoimmune diseases. Finally, how the in vitro and in vivo interference of the glycolysis pathway successfully prevents the activation of these cells is described.

## 2. Metabolic Reprogramming of Macrophages, Dendritic Cells, and Neutrophils

### 2.1. Macrophages

Monocytes (Mo) and macrophages (Mϕ) are cellular subsets in innate immunity, in defense against pathogens and in tissue repair. However, in case of an immune homeostasis imbalance, they initiate and regulate the autoimmune disease and other immune-based responses [12,13]. M1 and M2 represent two types of Mϕ; the M1 cells, having pro-inflammatory roles, are stimulated by IFN-γ or Toll-like receptor (TLR) ligands such as LPS; while M2 Mϕs are anti-inflammatory and are induced by IL-4/IL-13 [14,15]. Here, we will discuss the metabolic pathway of M1 Mϕs, as a critical player in some autoimmune disease progression.

Two primary states have been observed for macrophages, known as the quiescent and activated states. In the former, M1 cells exploit OXPHOS to produce sufficient ATP to meet basic cellular survival demands, while in the latter, aerobic glycolysis is preferred. Following stimulation, M1 cells rapidly adjust their metabolic strategy to glycolysis [16], which is induced by two metabolic regulators: mammalian target of rapamycin (mTOR), and Hypoxia-inducible factor-1 (HIF-1). mTOR activation enhances the transcription of HIF-1α mRNA, which is a crucial transcriptional pathway and downstream of various metabolic and immune signals [17,18]. HIF-1α significantly impacts the cellular metabolism. It increases the expression of many glycolytic enzymes, including HK2, triose-phosphate isomerase (TPI), and glyceraldehyde-3-phosphate dehydrogenase (GAPDH), leading to upregulation of the overall rate of glycolysis [18,19]. Moreover, it increases the expression of the glucose transporter (GLUT1 and GLUT3), to uptake more glucose [20,21]. Moreover, HIF-1α, by affecting pyruvate dehydrogenase kinase 1 (PDK1), results in the inhibition of pyruvate dehydrogenase (PDH). PDH is the enzyme that converts pyruvate to acetyl-CoA, allowing it to enter the Krebs cycle. Therefore, inhibition of PDH by PDK1 suppresses the Krebs cycle and mitochondrial activity [22]. Another target of HIF-1α is LDHA, which becomes upregulated after HIF-1α activation and induces the conversion of pyruvate to lactate and the monocarboxylate transporter 4 (MCT-4) (lactate transporter) [23,24]. Mitochondria dysfunction, which results from HIF activity, occurs following Max interactor 1 (MXI1) and Cytochrome c oxidase subunit 4 (COX4) inhibition. Concurrently with upregulated glycolysis, ATP production via the mitochondrial electron transport chain (through the HIF-dependent repression of genes that promote oxidative phosphorylation) is impaired [25,26].

Therefore, at the end of glycolysis, pyruvate encounters mitochondrial dysfunction, an impaired Krebs cycle, and an increased level of LDHA, which leads to the inability of pyruvate to enter into the Krebs cycle, and it is then reduced to lactate by LDHA. Excess lactate provides a feed-forward signal for glycolysis to proceed uninhibited, inducing the synthesis of inflammatory cytokines and mediating the development of the Th17 cell, which in turn exacerbates the inflammatory response [27,28].

In parallel with those events, the enhanced pentose phosphate pathway (PPP) also boosts the production of NADPH, to produce reactive oxygen species (ROSs) by NADPH oxidase (Figure 1) [29,30]. ROS induces translocation of the cytoplasmic enzyme pyruvate kinase M2 (PKM2) to the nucleus, where it phosphorylates STAT3, activating the production of IL-1β and IL-6 [31,32,33]. Those cytokines are accompanied by accumulated lactate and citrate to elicit inflammatory responses [34]. Thus, the metabolic pathways significantly affect immune responses and can lead to a pathogenic process.

M1 macrophages adjust their metabolic strategy to glycolysis following stimulation, induced by mTOR/HIF-1α signaling (not shown). HIF-1α activation (not shown) results in upregulation of GLUT, lactate transporter, and LDHA. Those events are coincidental with the PPP cycle, impaired Krebs cycle, and dysfunction of the electron transporter chain. Therefore, pyruvate is converted to lactate by LDHA, which elicits inflammation (GLUT (Glucose transporter), LDHA (Lactate Dehydrogenase A), PPP (pentose phosphate pathway), MCT (monocarboxylate transporters), MPC (Mitochondrial pyruvate carrier)).

### 2.2. Dendritic Cells

Besides Mo and Mϕ, the other important innate immune cells are the dendritic cells (DC). DCs are known for their function in antigen presentation and initiation of T cell response [35,36]. CD1c^+^ myeloid dendritic cells (mDCs) and plasmacytoid dendritic cells (pDCs) are the two major subsets of naturally circulating DCs (pDC) [37,38]. Contrary to pDC, mDCs are characterized by high expression of MHC-II, and they possess the integrin, CD11c [36]. Two major subsets of mDCs, according to their cell surface expression of CD8, CD11b, CD24, CD103, and Sirpα, have been identified [39]. Indeed, CD11b^+^ CD103^−^ and CD103^+^CD11b^−^ have been detected in peripheral tissues, while CD11b^+^CD103^+^ are localized in the gut [40,41].

DCs express a wide range of chemokine receptors. They are poorly immunogenic under homeostatic conditions with low expression of MHC molecules and co-stimulatory receptors, which could be the reason for their inability to stimulate T-cells sufficiently [42]. In the presence of danger signals and with stimulation of TLRs by inflammatory stimuli or pathogens, they maturate and secrete a high level of inflammatory cytokines, such as IL-8 and IL-6 [43,44,45]. In the last decade, it has been shown that DCs are essential players in many autoimmune diseases, making them a therapeutic target. With stimulation, DCs maturate and switch OXPHOS to glycolysis, with the effect of mTORC1 signaling and impact T cell activation [17,46]. Upon an initial inflammatory stimuli, the glycolytic pathway increases in DCs with a maintained OXPHOS, but they completely shift to glycolysis in a long-term activation [47]. In general, when activated immune cells use aerobic glycolysis, the regulatory and anti-inflammatory cells use OXPHOS [48]. Malinarich et al. demonstrated in 2015 that a high expression of many of the genes of the OXPHOS pathway, a higher content of proteins involved in this pathway, and increased mitochondrial activity were detected in tolerogenic dendritic cells (tolDCs), which contrasts with activated DCs. They showed that tolDCs generated after treatment, either with vitD3 alone or with dexamethasone and vitD3, had improved mitochondrial respiration and fatty acid oxidation (FAO) activity [49]. Accordingly, Ritprajak et al. concluded that the lipid accumulation that fuels FAO, skews DCs toward a tolerogenic function [50]. Hence, the importance of metabolic reprogramming in these cells, which can even lead a pathway to be pathogenic, could make DCs a therapeutic target.

### 2.3. Neutrophils

It is undeniable that changed lipids have a pro-inflammatory effect; additionally, many autoimmune disorders feature neutrophils in their pathophysiology. Increased lipid peroxidation in cell membranes in systemic lupus erythematosus (SLE) can cause lipid-derived reactive aldehydes (LDRA), such as malondialdehyde (MDA), phosphorylcholine, and 4-hydroxynonenal (4-HNE), to form, which can alter proteins and increase their immunogenicity [51,52]. It has been found that LDRA-specific immune complexes correlate with disease activity in SLE [53], implying that this histone H2AX phosphorylation can be used to assess the cellular damage associated with severe exposure to oxidative stress and other environmental variables. Many mitochondria-related effects, such as ROS synthesis, enlargement, polarization, enzyme activity, and apoptosis parameters, were modified in SLE [54]. For example, increasing oxidative stress and high levels of H2AX phosphorylation, identified in peripheral blood cell subsets, are correlated with SLE disease activity and consequently increase tissue damage and the capacity for cardiovascular damage in patients [55]. Oxidative stress is an imbalance between ROS production and the cell’s antioxidant capacity. This could be linked with an augmented risk of developing SLE and other autoimmune diseases [56]. A disruption in the NOX2 pathway caused by a missense mutation in the p47phox (NCF1) subunit of NOX could be an example of this imbalance. In the opposite situation, increased NCF1 copy numbers, associated with increased NOX2-derived ROSs, appear to protect against SLE [57]. Ribonucleoprotein–immune complexes (RNP ICs), which induce NETosis, require mitochondrial ROS for maximal neutrophils extracellular traps (NET) stimulation, resulting in mitochondria hyperpolarization and the release of NETs enriched in oxidized mtDNA [58]. Indeed, SLE NETs activate pDCs to produce high levels of IFN-α in a DNA- and TLR-9-dependent manner. Neutrophils do not only contribute to IFN-α production in SLE patients. They also secrete type I IFN by linking nucleic acid-recognizing antibodies in SLE patients [59]. Mitochondrial DNA (mtDNA) contains CpG oligonucleotides similar to bacterial DNA and could be an inflammatory agent [60]. mtDNA activation of neutrophils through TLR-9 [61] leads to cell-autonomous NOD-leucine-rich region- and pyrin domain-containing-3 (NLRP3) and TLR-9 activation [62,63], and could be a response to TLR-7 agonistic autoantibodies [64]. This type of neutrophil activation inhibits the disassembly of mtDNA–transcription factor A mitochondria (TFAM) complexes. TFAM is necessary for transferring oxidized (Ox) mtDNA into lysosomes for degradation, resulting in their retention within the mitochondria [64]. Consequently, exposure to IFN-α/RNP destroys the intracellular deposit of Ox mitochondrial products and leads to intramitochondrial accumulation and extracellular release of highly interferogenic Ox mtDNA bound to TFAM [64]. An in vitro analysis of neutrophils in SLE patients revealed that these cells exhibited mitochondrial modifications, including aggregation of Ox mtDNA, in a significant number of patients [64]. These modifications are not present in healthy children or patients affected by autoimmune diabetes, who also display upregulation of IFN-inducible transcripts, but do not carry the same TLR-7 agonist specificities [65]. Furthermore, Ox mtDNA induces an immune response in SLE, as autoantibodies against this modified DNA are detected in the patient sera [64].

mTOR pathway modulators and glucose metabolism inhibitors have demonstrated therapeutic benefit in experimental models of SLE, and rapamycin has also been tested in animal models of SLE and human rheumatic diseases [66,67,68,69]. For example, rapamycin inhibits disease activity in SLE patients and induces normalization of T cell activation, due to calcium flux [68]. The MYD88 scaffold protein also phosphorylates mTOR, leading to type-1 IFN production by IRF-5 and IRF-7 activation [70]. As previously stated, type I IFN secretion in SLE patients is mediated by a link between nucleic acid recognizing antibodies. This pathway is required for TLR induced type-1 IFN and autoantibody production by pDCs and B cells [71,72]. TLR mediated phosphorylation of mTOR and 4EBP-1 was inhibited in both pDCs and B cells by Gal-9 treatment, suggesting that activation of CD44 via Gal-9 may have played a role in this inhibition, associated with decreased recruitment of mTOR to the TLR-MYD88 complex and inhibition of TLR-mediated IRF-5 and IRF-7 activation [73].

## 3. The Possible Role of Mϕs, DCs, and Neutrophils in Autoimmune Diseases

The role of Mϕs, DCs, and neutrophils in autoimmune disease pathogenesis has been studied for many years. Nevertheless, most studies focused on their defensive role, while their pathologic role should also be considered.

### 3.1. Macrophages

As mentioned, macrophages show dynamic polarized phenotypes between M1 and M2, thanks to the action of pro-inflammatory and anti-inflammatory cytokines [74]. One of the most critical cytokines in the pathogenesis of many autoimmune diseases is IFN-γ, which stimulates macrophages to polarize into the M1 subset [75]. This polarization consequence in a strong pro-inflammatory phenotype is defined by the production of inflammatory cytokines such as TNF-α, IL-6, IL-12, and IL-23, chemokines such as CCL-5, CXCL9, CXCL10, and CXCL5, and the recruitment of Th1 and NK cell promotion. Moreover, M1 Mϕ upregulates the intracellular expression of a protein named suppressor of cytokine signalling 3 (SOCS3), which increases the production of ROSs and nitrogen, the expression of MHC class II molecules, and costimulatory molecules [76,77]. Thus, M1 Mϕs not only promote the Th1 immune responses but also induce tissue destruction [78]. Such tissue destruction occurs in various inflammatory, autoimmune, and chronic diseases, such as Crohn’s disease (CD), rheumatoid arthritis (RA), diabetes, multiple sclerosis (MS), and autoimmune hepatitis [78,79,80,81,82,83,84].

SLE is an autoimmune disease characterized by antibodies to nuclear and cytoplasmic antigens. A role for Mϕ in the pathogenesis of SLE was first proposed after it was found that SLE macrophages were unable to clear apoptotic cell debris [85]. Moreover, the macrophage activation signature between active and inactive SLE patients has been studied by hierarchical clustering of differentially expressed (DE) genes [86]. It was revealed that, even though M1-associated genes were upregulated in both active and inactive SLE patients, M2-associated genes tended to be more upregulated in inactive SLE patients [87]. Biologically informed gene clustering (BIG-C) unique to each cohort confirmed an effector function upregulation in Mo derived from active SLE [87]. In another study, they examined the effect of macrophage depletion in a pharmacological induced-lupus experimental model, and they found an increased SLE severity [88]. Interestingly, transferring M2 Mϕs reduced SLE severity, while transferring M1 Mϕs increased disease activity [89]. In any case, the problem could not be solved with complete Mϕs depletion.

In RA, patients endure increasing cartilage and bone destruction, and they experience remission, even after taking biological antirheumatic drugs such as TNF-α-inhibitors [47,90]. Some studies on both the synovial tissues and fluid of patients with RA indicated a high expression of proinflammatory genes and confirmed that M1 Mϕs are the dominant cells [34]. They contribute either as a significant cytokine producer or as antigen-presenting cells (APCs), to stimulate T cells and promote joint destruction [91,92,93]. Recently, distinct RA synovial macrophages have been identified in RA synovial. Four Mo/Mϕ populations were defined as TLR-activated IL-1β, which were expanded in synovial and produced pro-inflammatory cytokines [94]. The Mϕs in RA uptake a high glucose level by overexpressing GLUT1, leading to a significant volume of ATP production, which puts great stress on the mitochondria to leak ROSs [95]. In line with high infiltration of Mϕs, increased glucose uptake, and cytokine production, the total Mϕs were depleted in animal models of RA, resulting in a more favorable prognosis, reduced inflammation, and decreased joint destruction [96]. Macrophage polarization in the peripheral blood of RA patients was identified with highly expressed pro-inflammatory factors, such as HLA-DR, CD86, CD64, and CCR5 [47,97]. An increased number of M1 Mϕs in synovium leads to the secretion of chemokine C-X-C motif ligands (CXCLs) and promotes inflammation and angiogenesis from CXCL1 to CXCL10 [98]. In the synovial fluid and serum of RA patients, a high level of CXCL8 is associated with clinical manifestations [98,99]. The increased CXCL16 level in the synovium is implicated in monocyte recruitment into the synovial tissue [100]. In atherosclerotic lesions, Mϕs are continuously exposed to accumulating lipids and their oxidized derivatives [101]. It seems that cholesterol crystals deposited in the early stages of the atherosclerotic lesions’ development are responsible for the inflammatory activation of macrophages [102]. Oxidized lipoproteins promote inflammation, either by stimulating the caspase1 activating NLRP3 (cryopyrin) inflammasome to cleavage and secretion of IL-1 [103], or by a mechanism that involves inhibition of the transcription factor Kruppel-like factor 2 [104,105]. Additionally, minimally oxidized LDL promotes an inflammatory macrophage phenotype through activating a TLR-4 mediated pathway [106]. Regarding this, effective therapeutic strategies targeting several cytokines also affect the Mo/Mϕ axis [107]. Therefore, infiltrated Mϕs could be used as biomarkers for disease activity and a predictor of responsiveness to disease modifying antirheumatic drug (DMARD) therapy [108]. In the light of new approaches, immunity metabolite dysregulation could be recognized many days before clinical signs appear [109]. Luckily, fluorodeoxyglucose (FDG-PET) identifies the increased glucose uptake in RA patients early on, which correlates with disease severity and treatment response [110].

Mϕs’ pathologic roles have also been identified in autoimmune encephalomyelitis (EAE) and multiple sclerosis (MS). M1 Mϕs are now considered a crucial step in the pathological cascade of MS and EAE, through upregulation of the expression of major histocompatibility complex class II (MHC-II), scavenger receptors, and CD40 [111,112], releasing TNF-α, IL-1β, IL-6, IL-12, and IL-23, as well as chemokines, including CCL4, CCL5, CCL8, CXCL9, CXCL10, CXCL2, and CXCL4, as well as a low level of IL-10 [113,114]. A metabolic switch within macrophages intricately regulates their activation states. A recent experimental model study detected a high level of LDHA, MCT-4, and HIF-1α, corresponding with CD68^+^ Mϕs in perivascular cuffs of postcapillary venules, which is one of the main routes for leukocytes such as T cells and macrophages [28].

However, a skewing of the Mϕs phenotype toward M2 has been noticed in some diseases, but it seems this method is ineffective in CD. In both ulcerative colitis and CD, a perturbation in macrophage metabolic homeostatic pathways might significantly affect macrophage function and disease progression [115]. Since M2 Mϕs are potential fibrotic aggravators and cause granuloma formation in CD, any new medications that can target and inhibit both M1/M2 pathways in immune bowel disease (IBD) should be evaluated to restore macrophages’ ability to maintain immunological homeostasis in the gut lining and combat xenobiotics [115].

Stimulation of cultured human primary macrophages has also been performed to clarify these cells’ roles. For instance, stimulation with hyperglycemia (HG) revealed the acute production of TNF-α and long-term production of IL-1β during macrophage differentiation [116]. Chronic HG causes an abnormal metabolic environment, leading to diabetic complications [117]. M1 Mϕs activate T cells via expressing costimulatory molecules such as MHC, CD40, and CD86 and, through the expression of iNOS and ROS, produce proteases such as NO [118]. Consequently, ROSs activate the caspase pathway and induce excessive cell stress, mediating apoptosis or necrosis of insulin-secreting β cells [118,119]. The serum samples of pre-diabetic patients and HG treated human monocyte-derived macrophages also showed a lower IL-10 concentration [120]. Mouse peritoneal macrophages treated with the hyperglycemic medium were also investigated for cytokine production and an increase in the expression of pro-inflammatory cytokines, including IL-1β, IL-6, IL-12, IL-18, and TNF-α, in a time and dose-dependent manner was found, where the induced T1D cells increased IL-12 expression via activation of the protein kinase C (PKC), p38 MAPK, and JNK pathways [121]. Increasing TLR expression, such as TLR-7 and TLR-10 in the adipose tissue (AT) of obese individuals, is associated with increased expression of inflammatory markers in AT [122,123]. In obese individuals, the AT local proinflammatory microenvironment, reduction of adiponectin production, and, of note, upregulation of GLUT1 drive activation of adipocytes and skew Mϕs differentiation toward a M1-like phenotype [124,125,126,127]. These events coincided with the oxidative stress and mitochondrial dysfunction induced by HG, which further supports ROS production to activate the NFκB and p38 MAPK signaling pathways, glycolytic metabolism, and release of pro-inflammatory mediators by macrophages [128,129]. Hence, regarding the T1D induced by M1 Mϕ and the ability of M2 Mϕ in the production of IL-4/IL-13, experimental adoptive transfer of immunomodulatory M2 cells in NOD mice were reported to prevent T1D [119,130].

### 3.2. Dendritic Cells

In addition to Mϕs, there are signs of DC presence in many diseases. They display a tolerogenic and an immature phenotype in microbial-rich environments, such as the gastrointestinal, and are characterized by low expression of pattern recognition receptors (PRRs), including TLR molecules, and lower expression of both MHC-II molecules and surface costimulatory molecules [131,132]. In some studies, it was reported that some of the properties of to these cells are altered during IBD. For instance, an increase in their number in the inflamed mucosa, a reduction of the CD103^+^ subset, and an elevated expression of PRRs are caused by these cells, which lead to losing tolerogenic function, providing a higher capacity to recognize microbial antigens and a pro-inflammatory profile [131,133,134,135]. Although the overlapping in functions, activities, and lineage markers of DCs with mononuclear phagocytes makes them hard to distinguish and leads to miscellaneous conclusions [136], assessing the levels and/or properties of circulating DC could be a helpful approach to help in monitoring such patients [133,137].

Emerging evidence suggests a role for DCs within the adipose tissue and chronic inflammation. Rather than adipose tissue macrophages (ATMs), adipose tissue DCs (ATDCs) are more APC, whose MHC-II signals are required for CD4^+^ cell maintenance [138]. While how exactly adipose tissue DCs, and, if so, which subsets, contribute to inflammation in obesity remains incompletely understood, a recent study suggests that adipose tissue CD11b^+^ CD103^‒^ DC expressed more IL-1β, IL-6, and IL-23 mRNA at the transcriptomic level and can affect Th17 maturation [36]. As obesity is more characterized by Th1 inflammation, the minor group CD103^+^ CD11b^‒^ DCs that express more IL-12 and IL-18 are a well-established player in diabetes development [36,139]. Besides diabetes, IBD, and SLE, migration of DC to the central nervous system (CNS) is considered a critical event in the pathogenesis of MS, whose increasing number during both EAE and MS is correlated with the disease’s activity [140,141].

In early studies, pDCs were first described as the main producers of IFN-α after viral infection [142]. A prominent IFN-α signature was also reported in SLE, related to active disease [143]. Then, pDCs were identified as the essential source of IFN-α in SLE patients [144]. Autoantibodies, DNA/RNA, and virus- or bacteria-derived nucleic acids stimulate pDCs to produce IFN-α through TLR-7/TLR-9 signaling [56,145,146,147,148]. MRL/Mp-Faslpr (lpr) lupus mouse models were used to detect IFN-α production. It was observed that pDCs from the late-stage disease could not produce IFN-α, while IFN-α was produced by pDCs mainly at the early stages of the disease [149]. Another study by Farkas et al. showed that SLE pDCs were diminished in the blood of SLE patients. However, they were accumulated in lupus patients’ inflamed or damaged skin, suggesting their rapid migration from blood to the inflamed tissues [150]. The use of different lupus mouse models, such as BXSB.DTR mice has revealed that early activity of pDCs contributes to inducing SLE disease symptoms through the activation of T cells, B cells, and cDCs. The same group reported that early and transient depletion of pDCs could ameliorate autoimmunity in a lupus model [151].

### 3.3. Neutrophils

Following the well-studied pathologic role of Mϕs and DCs, the most common immune cell, neutrophils, and their ability to be swiftly recruited to infection or damaged tissue, if activated persistently, could generate destructive effects, initiating the auto-immune process. Such a pathological role of these cells has been seen in the pathogenesis of T1D [152]. Neutrophils are involved in T1D pathogenesis through two mechanisms; first, they secret IL-17 to promote granulopoiesis, causing neutrophils’ proliferation and accumulation. Second, they mediate the entrance of bacteria in the pancreas that release toxins leading to IL-6 and CXCL-8 production that attracts more neutrophils [153]. In the study by Huang et al., the liberation of neutrophils’ toxic mediators and the anti-neutrophil treatments could ameliorate insulitis and autoimmune diabetes [154].

Besides the mentioned ability of neutrophils, many pathologies involved in autoimmunity are related to the action of the NETs components, which are considered notable players in IBD pathogenesis [155]. The gut epithelial barrier is damaged in the presence of inflammatory cytokines by neutrophils. Direct tissue damage in the gut can be caused by neutrophils, due to releasing proteases, such as matrix metalloproteases and neutrophil elastase (NE) and calprotectin [156,157]. MPO is the most abundant cytotoxic enzyme within the azurophilic granules of neutrophils and participates in the immune response through the activation of T cells [158]. Regarding the generation of NETs, and consequently, the high level of MPO in both plasma and synovia of RA patients, depletion of MPO was considered in a mouse model with arthritis, and the result was associated with a reduction of the disease severity [159,160]. In addition to MPO, activated neutrophils, especially infiltrated neutrophils in RA synovial fluid, show a high expression of anti-apoptotic protein Mcl-1. Therefore, they exacerbate inflammation by delaying the apoptosis process, which is reinforced by other molecules and conditions, such as ROSs, leukotriene B4, granulocyte-colony stimulating factor (G-CSF), granulocyte macrophage-colony stimulating factor (GM-CSF), IFN-γ, IL-1β, IL-15, TNF-α, and the local oxygen tensions within the joint [161,162,163,164]. Among these diseases, MPO and NE are increased in the plasma of MS and EAE patients; thus, MPO depletion and NE inhibition were examined in EAE and found to decrease disease severity [165,166].

## 4. Interfering with the Glycolytic Pathway: In Vitro and In Vivo Models

Due to autoimmune disease’s increasing prevalence and regarding the fact that immune metabolites could be recognized much earlier than symptom onset, detection of increased GLUT1 and high glycolysis-related genes and, therefore, targeting OXPHOS, mTOR, and/or glycolysis, could be considered a therapeutic approach to minimize the side-effects of the traditional immunosuppressant drugs.

Inhibiting glycolysis in a specific type of cells could be an exciting technique. Several nonprotein nutrient kinases can modulate glycolytic energy flux and impact macrophage activation. For example, carbohydrate kinase-like protein (CARKL, a carbohydrate kinase whose substrate is sedoheptulose) is a repressor of the NADH: NAD^+^ ratio in LPS-induced murine models macrophage activation and antagonizes the production of NF-κB-regulated cytokines [167,168].

Blocking glycolysis in Mϕs in some infection diseases, such as severe sepsis, also diminishes severity. For example, in vitro inhibition of pyruvate kinase M2 (PKM2), which is the regulator of the rate-limiting step in glycolysis, with Shikonin can attenuate NLRP3 and absent in melanoma 2 (AIM2) inflammasome activation and, thus, suppress IL-1 release, as confirmed by conditional PKM2 knockout [169,170].

In vivo experiments show that blocking of glycolysis through a small molecule named 3-(3-pyridinyl)-1-(4-pyridinyl)-2-propen-1-one (3PO) can antagonize PFKFB3 (6-Phosphofructo-2-Kinase/Fructose-2,6-Biphosphatase 3). Regarding the role of PFKFB3 in glycolysis, its inhibition could reduce glucose uptake and can weaken sepsis-related acute lung injury, by suppressing inflammation and apoptosis of alveolar epithelial cells [171]. Based on these studies on glycolysis inhibition, Meng et al. showed that in vitro blocking of glycolysis with Dexmedetomidine (DEX), a selective α2-adrenoceptor agonist, inhibits the uptake of glucose and lactate production, as well as the production of inflammatory cytokines, via suppressing HIF-1α activation, in part, by inhibiting mTOR activation. Therefore, the same group suggested that DEX might weaken the LPS-induced inflammatory responses and glycolysis in activated macrophages [172].

Interference with the glycolytic activities in macrophages was also indicated by Kaushik et al., who showed a high expression of LDHA within the perivascular cuff of postcapillary venules. They concluded that in addition to blocking MCT-4 functions, inhibition of glycolytic through LDHA inhibitor (3-dihydroxy-6-methyl-7-(phenylmethyl)-4-propylnaphthalene-1-carboxylic acid (FX11) decreases both glycolysis (indicated by lactate production following glucose addition) and glycolytic capacity (the maximum capacity of lactate generated after inhibition of oxidative phosphorylation) on LPS-stimulated macrophages. Hence, they reported that hindering glycolysis is an attractive therapy to reduce and modulate the severity of inflammatory conditions, such as EAE disease and MS; and the functional relevance of glycolysis was confirmed by siRNA-mediated knockdown of LDHA and MCT-4 [28].

Another inhibitor of M1 polarization is the soluble death receptor 5-Fc (sDR5-Fc) fusion protein. Guang et al. administered sDR5-Fc to murine macrophage cell line RAW264.7 cells under various conditions, including nutrition deficiency, excessive peroxide, and ultraviolet irradiation, to explore the effects of sDR5-Fc. As a result, they discovered that administration of sDR5-Fc can inhibit the expression of HK2 and GLUT1 mRNA and protein and the expression of IL-1, TNF-α, mRNA, and proteins [173]. sDR5-Fc is a DR5 extracellular region structure that suppresses DR5 and blocks a series of subsequent signal transduction processes by competitively combining the tumor necrosis factor-related apoptosis-inducing ligand (TRAIL) with DR5 on the cell membrane surface; thus, protecting cells from apoptosis [173,174]. Considering that the interaction between TRAIL and TRAIL-R could contribute to some autoimmunity processes such as SLE [175,176], blocking the interaction between endogenous TRAIL and DR5 through sDR5-Fc might reduce the glycolysis rate of M1 Mϕs.

In vitro experiments on pre-treated monocyte-derived DCs have shown that FP7 (a small-molecule synthetic, TLR-4 antagonist) selectively antagonizes TLR-4 induced metabolic reprogramming in DCs and monocytes in a concentration-dependent manner, which leads to inhibition of glucose consumption and prevents the induction of cytokine secretion [44]. One of the upregulated genes of glycolysis-related genes is *ENO2* (encodes a dimeric enzyme, Enolase). *ENO2* catalyzes the second to last step in glycolysis, i.e., interconverting 2-phosphoglycerate (2-PGA) and phosphoenolpyruvate (PEP). Inhibiting *ENO2* using SF2312 in in vitro experiments revealed an impairment in the maturation and activation of myeloid DCs [177,178]. Recently, clinical trials have reported that the generation of tolDCs could be a therapeutic approach in some autoimmune diseases. TolDCs induce immune tolerance by promoting T cells hyporesponsiveness [179], preventing autoimmunity by triggering regulatory T cell (Treg) development, and inhibition of effector and autoreactive T cells [50,180,181]. Such a study showed that a generation of tol-DCs with different immunosuppressive agents, such as vitamin D3 or corticosteroids, could induce regulatory T cell function and effectively decrease the incidence of EAE and type 1 diabetes [182,183,184]. Vitamin D3 shifts activated DCs toward tolDC phenotypes in autoimmunity patients, and promotes Treg generation [185,186,187]. Additionally, intraperitoneal administration of autologous tolDCs in refractory Crohn’s patients revealed an increase in circulating T-regs and a decreased level of IFN-γ production after T-cell activation [188].

The neutrophil metabolism is much more complicated. Borella et al. showed that components related to NETs, such as myeloperoxidase, were increased in neutrophils from severe COVID-19 pneumonia patients. Neutrophils from those patients showed increased glycolysis, GLUT1, LDHA, and stabilized HIF-1a. Regarding the important role of HIF-1a in the regulation of NETosis, they treated neutrophils with Chetomin, and found that chetomin can inhibit HIF-1α [189], which is also the master regulator of glycolysis.

ROS also play a critical role in neutrophil-induced pathology. These cells use their mitochondria to generate ROS, and that mitochondrial ROS (mROS) is increased in hypoxic conditions, which are required for neutrophil survival. Therefore, Willson et al. suggest that in vitro inhibition of Glycerol-3-Phosphate Dehydrogenase 2 (GPD2) using the inhibitor iGP-1 was sufficient to accelerate neutrophil apoptosis and reduce mitochondrial membrane potential [190].

In addition to blocking either glycolysis or the factors involved in glycolysis in specific cells, we can see how blocking GLUT1, the enzymes, and glycolysis can effectively prevent some disease prognosis.

So far, glycolysis inhibitors such as 3-bromopyruvate (3-BrPA) and 2-deoxy-D-glucose (2DG) have been recognized which could decrease the available ATP [191,192]. HK phosphorylates 2DG in the first step of glycolysis, leading to the accumulation of 2DG-P (non-hydrolyzable substrate) in the cytoplasm and subsequent inhibition of both glycolysis and the pentose phosphate pathway. Thus, the inhibition of glycolysis by 2DG forces cells to use other mechanisms to generate ATP, such as reactivation of downregulated OXPHOS pathways [191]. The effects of 2DG have also been examined in a K/BxN mouse model of arthritis to inhibit glycolysis, which showed that 2DG significantly decreases joint inflammation, as well as adaptive and innate immune cell activation [193]. The preclinical result of 2DG administration in patients with metastatic or advanced solid tumors combined with weekly docetaxel showed that 2DG could be a good combination in many fields for inhibition of glycolysis [194].

In 2016, Wang et al. examined the impact of 3-BrPA and sodium citrate (SCT) to target glycolysis in gastric cancer cells, to investigate the effects of inhibitors on glycolysis. They discovered that 3-BrPA and SCT could target glycolysis by decreasing lactate and ATP production in nude mice in an in vitro experiment and in vivo stomach orthotopic transplantation tumor growth. Interestingly, the same team discovered that 3-BrPA significantly reduces the activity of the glycolytic enzyme HK. At the same time, SCT selectively inhibits the action of the glycolytic enzyme PFK-1 in a time and dose-dependent manner [195].

Glycolysis affects the pathologic progress of acute and chronic inflammatory diseases, such as acute lung injury (ALI). Zhong et al. explored the effects of glycolysis in an animal model of ALI and showed that glycolysis is enhanced during the ALI. Thus, they reported that blocking glycolysis with 2-DG suppresses the activation of NLRP3 inflammasome in vitro and attenuated ALI in a murine model induced by LPS in vivo. As such, they suggested glycolytic inhibition as an effective anti-inflammatory strategy in treating ALI [196]. Inhibition of glycolysis by 2-DG was also identified in the prevention of autoimmune activation in treating lupus-prone mice and reducing joint inflammation in a mouse model of arthritis [193,197].

While there are no such clinically tested molecules and components in the inhibition of glycolysis, some molecules, such as STF-31 and WZB117, have been explored in the inhibition of GLUT1 in some cancers [7].

Considering the role of GLUT1 as a master factor in initiating glycolysis, its targeting could be critical in preventing diseases. Not many specific GLUT1 inhibitors have been examined, although some molecules such as STF-31 and WZB117 have been explored in the inhibition of glycolysis in some cancers [7]. Daily intraperitoneal injection of a soluble analog of STF-31 has been reported to reduce of growth of tumors of von Hippel-Lindau (VHL)-deficient cancer cells grafted on nude mice [198,199]. Chan et al. showed that treatment of VHL-deficient cells with STF-31 leads to inhibition of glucose uptake on glycolysis, and in a dose-dependent manner on the production of ATP [198].

WZB117 is another GLUT1 inhibitor, which has been examined in lung cancer A549 and breast cancer MCF7 cells by Liu et al. They showed that WZB117 induces inhibition of cell growth, especially in hypoxic conditions [199]. Daily intraperitoneal injection of WZB117 in mice tumor models showed a significantly smaller size of compound-treated tumors. The same group concluded that in vitro and in vivo treatment with WZB117 results in a reduction of glucose transport and also in some glycolysis enzymes and metabolites [199].

Small clinical trials have proved that using rapamycin (single daily oral administration) inhibits mTOR in SLE, where it can normalize T cells’ activation and diminish disease activity [68]. Even though some side effects have been reported during the therapy with rapamycin in SLE patients, it has been identified as a safe medication [200]. Rapamycin, in combination with N-acetylcysteine (NAC), in SLE patients, causes a decrease in oxidative stress through inhibition of mTOR. [201,202]. The therapeutic features of rapamycin have also been reported in RA and Sjogren’s syndrome (through increasing of Treg cells) [203,204] and also in a phase I trial of systemic sclerosis patients [205].

Regarding in vitro experiments, that showed metformin (Met) is an inhibitor of mitochondrial metabolism [206] and also a regulator of glucose, Met is used in the treatment of type 2 diabetes mellitus (T2DM) [207,208]. Based on the effects of Met, Yin et al. examined the impact of Met (an AMPK activator) in combination with 2DG in TC and NZB/W lupus mice models. They found that these components decreased either oxygen uptake by mitochondrial T cells or mTORC1 signaling and glycolysis, which has been discussed by Stathopoulou et al. [201,206]. The immunomodulatory drugs used in SLE and lupus nephritis, Mycophenolate mofetil (MMF), exert their effects through inhibition of the PI3K/Akt/mTOR axis in T cells. Methotrexate is a widely-used drug in autoimmunity diseases, such as RA, and inhibits dihydrofolate reductase (DHFR) and decreases glycolysis and ATP levels, and so is used in RA, CD, psoriasis [207], and MS [209].

Dimethyl fumarate (DMF) is used in the treatment of MS and psoriasis, and its metabolites (monomethyl fumarate) succinate the glycolytic enzyme glyceraldehyde-3-phosphate dehydrogenase (GAPDH), resulting in decreasing glycolysis and increasing OXPHOS [201,210,211,212]. DMF has immunomodulatory effects on a wide range of cells. In macrophages, it promotes reprogramming of M1 cells toward M2 cells [207], and in DCs decreases either expression of pro-inflammatory cytokines induced by LPS or pro-inflammatory micro-RNA miR-155 expression [207,213,214,215]. A brief description of interventions is shown in Table 1.

Hence, any in vitro cell culture that targets Mϕs and DC metabolic reprogramming, primarily through inhibition of either master regulators, such as HK, LDHA, and glucose uptake, to effect glycolysis or to block mTOR (Figure 2), and then with measurement of the metabolites through laboratory techniques, could be considered before in vivo experiments. Thus, the generation of knockout models for the genes involved in maintained pathways could be an exciting approach to preventing various diseases.

mTOR is a serine-threonine kinase that acts downstream of the phosphoinositide-3-kinase (PI3K) and AKT signaling cascade [216]. Following cell stimulation, activated mTOR through HIF-1α induces the expression of the genes (not shown) needed for glycolysis to support the high energy demand. Although this activation is a defense line, in the case of stimulation with autoantigens, it can induce autoimmune disease. Therefore, the inhibition of the two main actors, glycolysis and/or mTOR, could be a therapeutic approach to disease progression.

## 5. Conclusions

All these data are intended to demonstrate the importance of metabolic reprogramming in the innate immune cells. Glycolysis and OXPHOS are two critical pathways regulating metabolic reprogramming, while they are tightly controlled by enzymes and pathways. Therefore, if any factors, such as persistent stimulation of cells, interfere with the operation of these controllers, this can upset the balance of glycolysis/OXPHOS and cause more activation of cells to being pathologic. So far, the pathologic role of innate cells has been reported in many autoimmune diseases, such as SLE, RA, etc. This review suggests that at least two mechanisms involved in the autoreactivity could be exerted by metabolic reprogramming of innate immune cells: (i) direct activation of metabolic pathways, such as glycolysis and mTOR, that induce excessive immune responses, resolving in autoimmune disorders; (ii) an indirect mechanism acting on pDC via the immunocomplex-driven release of IFN-α induced autoimmune diseases.

Any agent that avoids glycolysis exacerbation could be exploited to inhibit excessive glycolysis in macrophages, dendritic cells, and neutrophils, considering the role of these cells in many pathologic conditions.

Nevertheless, while some immuno-modulatory drugs of the glycolysis/mTOR pathway, such as MMF, methotrexate, and DMF, are used in some autoimmune diseases, many potent glycolysis-inhibitory molecules are still in the pre-clinical phase of study [7]. 

## Figures and Tables

**Figure 1 cells-11-01663-f001:**
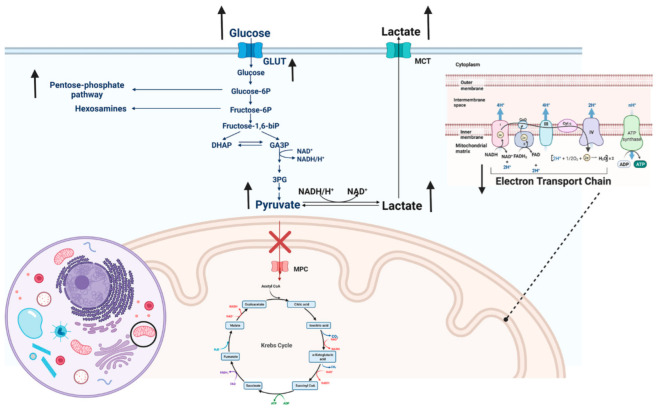
Downstream of HIF-1α activation. Created with BioRender.com (accessed on 22 April 2022).

**Figure 2 cells-11-01663-f002:**
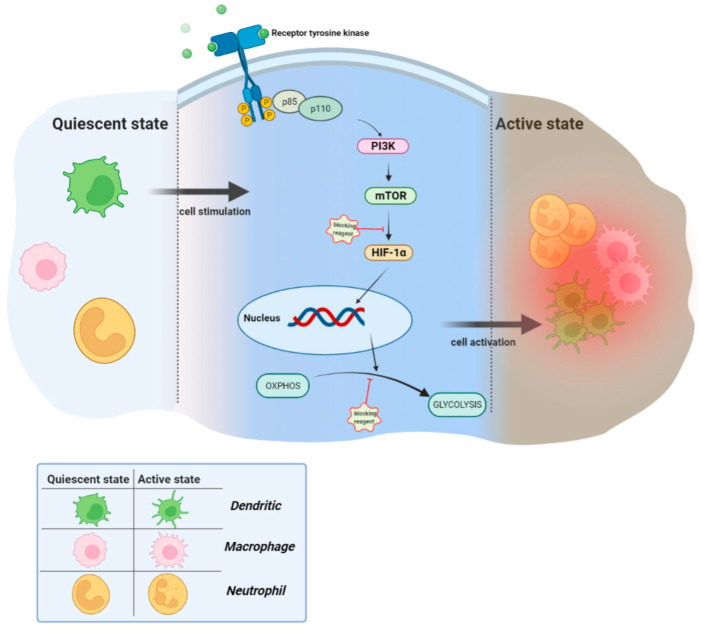
Inhibition of mTOR and/or OXPHOS-glycolysis switching. Created with BioRender.com (accesed on 22 April 2022).

**Table 1 cells-11-01663-t001:** A brief description of the molecules that affect metabolic reprogramming and the major metabolic pathways and/or enzymes that are affected by those factors.

Metabolic Modulators	Target
CARKL	PPP, NF-kB-regulated cytokines
Shikonin	Glycolysis (PKM2)
3PO	Glycolysis (PFKFB3)
DEX	Glycolysis, HIF-1α
FX11	LDHA
sDR5-Fc	HK2 and GLUT1
FP7	TLR-4
SF2312	ENO2
Chetomin	HIF-1α
2DG	Glycolysis
iGP-1	GPD2
3-BrPA	Glycolysis (HK)
SCT	Glycolysis (PFK-1)
STF-31 and WZB117	GLUT1
Rapamycin	PI3K/Akt/mTOR
NAC	mTORC1
Met + 2DG	mTOR/Glycolysis
Methotrexate	mTOR
DMF	Glycolysis (GAPDH)

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
