# Peer review of "Metabolic Reprogramming of Innate Immune Cells as a Possible Source of New Therapeutic Approaches in Autoimmunity"

_cells, 2022, doi:10.3390/cells11101663_

Round 1

Reviewer 1 Report

The article is consistent within itself. The references are relevant and recent. The cited sources are referenced correctly. Appropriate and key studies are included. The paper is comprehensive, the flow is logical and the data is presented critically. The quality of figures is outstanding.

Specific comments on weaknesses of the article and what could be improved:

Major points  

  1. The structure is a bit elusive - many sections that state "role of ... cells in autoimmunity" followed by "reprogramming of these cells", but this structure is not equivalent for all the cell types. Could you consider an adjustment of sections and probably, merge some of them for a certain cell type?
  2. 7. Interfering with Glycolytic Pathway, in-vitro and in-vivo Models - Please, clarify what is interfering with this pathway
  3. too little is devoted to the treatment options, but they are stated in the title and the aim of the review. You can add a section for this, or update it throughout the text.

Minor points

  1. Could you please discuss the clinical implications of the results
  2. Update the keywords by adding terms such as "autoimmunity", "therapy", etc.

Author Response

We thank the editor and the reviewer for allowing us to improve our manuscript  by addressing their suggestions and comments.  We have reorganized the manuscript's structure. Moreover, we have better described the in vitro and in vivo experimental models to study the molecules that interfere with glycolytic pathways and, when studied, their possible application for clinical exploitation. Below is our point-by-point reply.

Reviewer 1:

  1. The structure is a bit elusive - many sections that state "role of ... cells in autoimmunity" followed by "reprogramming of these cells", but this structure is not equivalent for all the cell types. Could you consider an adjustment of sections and probably, merge some of them for a certain cell type?

Answer: We tried to make the manuscript more cohesive. After the introduction, we divided the review into three main parts. The first part: the metabolic pathway of macrophages, DCs, and neutrophils, is divided into three separate sections for macrophages, dendritic cells, and neutrophils. The same division has been performed for the second part: the possible role of macrophages, DCs, and neutrophils in autoimmunity. Finally, the third part is about the strategies for interfering with the metabolic pathways of innate immune cells, and it is composed of a single section.

  1. Interfering with Glycolytic Pathway, in-vitro and in-vivo Models - Please, clarify what is interfering with this pathway

Answer: In the part of ‘Interfering with Glycolytic Pathway,’ we described the function of molecules that can interfere with the activity of the enzymes involved in the metabolic pathways and their alteration, these molecules are also mentioned in the conclusions, and the most studied interfering molecules, are listed in table 1.

  1. too little is devoted to the treatment options, but they are stated in the title and the review's aim. You can add a section for this, or update it throughout the text.

Answer: Since there is not too much information about glycolysis inhibition in the clinical phase, especially about autoimmunity, we added some paragraphs in the part of Interfering with glycolytic pathway, in vitro and in vivo models, which initiates from line 359.

Minor revision:

  1. Could you please discuss the clinical implications of the results?

Answer: Based on major revision n.3, some paragraphs about the clinical implication have been added, which initiates from line 440.

  1. Update the keywords by adding terms such as "autoimmunity", "therapy", etc.

Answer: It has been performed.

Reviewer 2 Report

Mohammadnezhad et al. presented a review on the metabolic reprogramming  of innate immune cells as a possible source of new therapeutic approaches in autoimmunity. The authors reviewed literature on the metabolic reprogramming of macrophages, dendritic cells, and neutrophils as well as their connections to autoimmune pathogenesis. They also discussed the effect of interference with the glycolytic pathway in various immune models. This work is of broad interest to the field. I have the following suggestions to improve the readability of the manuscript:

  1. Please include an introduction section to help readers better understand the background and logic flow of the review.
  2. Please include a discussion/conclusion section to highlight the main conclusions and potential future directions.
  3. Fonts in Figure 1 and insert are too small to read.
  4. A figure or table summarizing main interventions on metabolic reprogramming, with connection to the major metabolic pathways discussed, will be helpful.
  5. Language editing will further improve the clarity of the manuscript.

Author Response

We thank the editor and the reviewer for allowing us to improve our manuscript by addressing their suggestions and comments.  We have reorganized the manuscript's structure and added an introduction and conclusions section. Moreover, we have improved the quality of the figure1 and added a table that summarizes the main molecules used to intervene in metabolic reprogramming. Below is our point-by-point reply.

Reviewer 2:

  1. Please include an introduction section to help readers better understand the background and logic flow of the review

Answer: We added the introduction section that briefly explains immunometabolism, glycolysis, and the aim of this review.

  1. Please include a discussion/conclusion section to highlight the main conclusions and potential future directions.

Answer: The conclusion part which consists of a future direction that can be performed in the field of in-vitro, in vivo, and animal models have been mentioned.

  1. Fonts in Figure 1 and insert are too small to read.

Answer: We modified the quality of figure1, so it’s more readable now.

  1. A figure or table summarizing main interventions on metabolic reprogramming, with connection to the major metabolic pathways discussed, will be helpful.

Answer: We summarized the main interventions, and which major metabolic pathways are affected by them, which you can find kindly in Table 1.

  1. Language editing will further improve the clarity of the manuscript.

Answer: We checked and corrected the manuscript for spelling and language.

Round 2

Reviewer 2 Report

Mohammadnezhad et al. presented a review on the metabolic reprogramming  of innate immune cells as a possible source of new therapeutic approaches in autoimmunity. The authors reviewed literature on the metabolic reprogramming of macrophages, dendritic cells, and neutrophils as well as their connections to autoimmune pathogenesis. They also discussed the effect of interference with the glycolytic pathway in various immune models. This work is of broad interest to the field. The authors have improved the readability of the manuscript and acceptance is recommended.

This manuscript is a resubmission of an earlier submission. The following is a list of the peer review reports and author responses from that submission.

Round 1

Reviewer 1 Report

Review of "Metabolic Reprogramming of Innate Immune Cells As a Possible Source of New Therapeutic Approaches in Autoimmunity" by Leila Mohammadnezhad et al.

The authors suggested that targeting metabolic reprogramming in innate immune cells, such as macrophages, dendritic cells, and neutrophils could reduce the disease severity of autoimmune diseases. They further indicated that the mTOR-HIF-1α and the OXPHOS-Glycolysis are the essential pathways. However, the review article shows no novelty. There are many studies showed that the metabolic reprogramming towards immune cells polarization (Cancers 2020, 12(6), 1411) (Immunometabolism. 2020;2(2):e200017) (Clin Exp Immunol. 2019;197(2):181-192). Moreover, Julianna et al have reported that the carbohydrate kinase-like (CARKL) protein is important in regulating macrophage metabolism which affects macrophage polarization (Cell Metab. 2012;15(6):793-5). Besides, there are some mistakes in the sentences, such as “andimmune” (line 41),” andGLUT1” (line 96), “theimmune” (line 97), “activatesIRF-5” (line 244), “couldplay” (line 248). In addition, some sentences should be checked for the format, for example, in line 39 (“…is favored. Although”), a space is required between the “endpoint” and “although”.

Please check the format and the grammar before submission. Due to these cumulative reasons, I recommend rejection of this manuscript in its current form.

Reviewer 2 Report

The authors describe all aspects of neutrophils and macrophages in autoimmunity, including the possible pathogenetic mechanisms.

However, as stated in the title " possible source of new therapeutic approaches", some paragraphs on these should be included.

If there are any in vitro or in vivo models, or clinical trials in this regard, they also should have been mentioned.

If there are not such, this should be stated in the paper. Authors also have to specualate what type of research is needed to develop therapy based on the knowledge described in this review.